# Can scholarly pirate libraries bridge the knowledge access gap? An empirical study on the structural conditions of book piracy in global and European academia

**Balázs Bodó**[1]*, **Dániel Antal**[2], **Zoltán Puha**[3]

**1** Institute for Information Law, University of Amsterdam, Amsterdam, The Netherlands, **2** Independent researcher, The Netherlands, **3** Department of Methodology and Statistics, Jheronimus Academy of Data Science, Tilburg University, Tilburg, The Netherlands

* bodo@uva.nl

**Data Availability Statement:** All data and code is publicly available: Code and data repository: https://zenodo.org/record/4012352; DOI: 10.5281/ZENODO.4012352 Raw Data repository: https://

## Abstract

Library Genesis is one of the oldest and largest illegal scholarly book collections online. Without the authorization of copyright holders, this shadow library hosts and makes more than 2 million scholarly publications, monographs, and textbooks available. This paper analyzes a set of weblogs of one of the Library Genesis mirrors, provided to us by one of the service's administrators. We reconstruct the social and economic factors that drive the global and European demand for illicit scholarly literature. In particular, we test if lower income regions can compensate for the shortcomings in legal access infrastructures by more intensive use of illicit open resources. We found that while richer regions are the most intensive users of shadow libraries, poorer regions face structural limitations that prevent them from fully capitalizing on freely accessible knowledge. We discuss these findings in the wider context of open access publishing, and point out that open access knowledge, if not met with proper knowledge absorption infrastructures, has limited usefulness in addressing knowledge access and production inequalities.

## Introduction

Library Genesis (LG or LibGen) is a copyright infringing online collection of scholarly works: monographs, edited volumes, and textbooks [1]. At the time of writing in May, 2019, there are 2,363,587 records in its online catalogue, accessible through a simple web interface. The digital versions of the books in LG are accessible via various centralized and peer-to-peer third-party services. All elements of the LG web service are freely available for anyone to download, including the webserver code, the most current copy of the database, or the works themselves.

LibGen contains several collections. Its main focus is scholarly works: scientific monographs, edited volumes, and textbooks. It also serves as a repository for scientific articles downloaded by the users of SciHub, another copyright infringing shadow library focused solely on journal articles [2], and a separate catalogue of literary work and comics.

uvaauas.figshare.com/projects/Shadow_Libraries/80837; DOI: 10.21942/uva.12330959.

**Funding:** The research received funding from the H2020 Research grant #710722 "OPENing UP new methods, indicators and tools for peer review, dissemination of research results, and impact measurement", and was carried out on the Dutch national e-infrastructure with the support of SURF Cooperative.

**Competing interests:** The authors have declared that no competing interests exist.

The legal status of LibGen is understood to be copyright infringement by rights holders, authors, as well as the users, and operators of the service [1]. A New York court issued a default judgement against SciHub and Library Genesis, including their operators, finding all liable for willful copyright infringement. The court ordered Alexandra Elbakyan, the operator of Sci-Hub, and the anonymous operators of LibGen to pay damages of $15M, as well as confiscating the domain names [3]. A Virginia district court ordered the domain names to be blocked in the US [4]. The domain names of the services were temporarily blocked in Russia, where these services were thought to be located [5], and by a number of ISPs in Europe. Online service providers such as Facebook have also filtered links to the service.

The administrators of LibGen (and SciHub, for that matter) did not contest the legal assessment [1, 6]. Neither did those who use these services by downloading or uploading materials from / to them [7]. Yet there seems to be a widely shared (but certainly not universal [8]) consensus in the academic sector about the moral acceptability of such radical open access practices [9–13]. Willful copyright infringement in the research and education sector is seen as an act of civil disobedience, resisting the business models in academic publishing that have faced substantial criticism in recent years for unsustainable prices and outstanding profit margins [14]. Since shadow libraries are a product of the cooperation between scholars, who contribute texts and other resources (such as donations, volunteer work, etc.), shadow libraries represent a 'bottom-up', radical approach to open access: a physical approximation of the Platonic ideal of knowledge sharing that would exist if there were no legal, economic, or institutional barriers to the circulation of scholarly knowledge.

The problematic nature of the current economic organization of scholarly publishing has long been acknowledged [7, 15–21]. The traditional model of academic publishing relies on access control, where publishers sell steeply priced subscriptions to journals, books to libraries, and textbooks to students. Its alternative, open access publishing, shifts the costs from readers to authors and their institutions by charging article processing fees to authors in exchange for free open access to their published articles. Both business models are exclusionary in one form or another. Access control regimes affected the least resourceful institutions first; in recent years, even the most financially well-endowed US Ivy league universities have warned about the unsustainability of subscription fees [22], or cancelled contracts with journal publishers [23]. In recent years, multiple institutional consortia and national science agencies in charge of agreements with academic publishers let them lapse in the hopes of reaching a financially more sustainable deal [24–29]. On the other hand, the article processing fees associated with the now standard Golden Open Access regimes create publication barriers for those researchers lacking institutional budgets to cover such costs.

Shadow libraries such as Library Genesis and SciHub were created in response to the complex institutional, political, financial, and economic conditions that limit access to knowledge at the geographic and institutional periphery of academia [1, 7, 9]. However, since these services are now deeply embedded in the current system of circulation of scholarly knowledge [2, 30], their current use is probably more complex than simply serving disadvantaged scholars, low-income countries, or underfinanced institutions.

There are very few empirical studies on the extent and potential impact of book piracy, in general, and scholarly piracy, in particular. There are many possible explanations for why online book piracy is rarely in the headlines: e-book markets and audiences are still relatively small compared to print; electronic reading device penetration is much lower than mp3 players and the like; and print is probably still a preferred format for many. Yet, while e-book piracy is definitely present, its volume and economic value is perceived as low, especially compared to the losses suffered by the music and audiovisual sectors [31]. E-book black markets failed to develop their own Napster service, and book piracy sites have remained local,

fragmented and marginal. As a result, it is challenging to study the supply and demand of these illicit services. The few existing studies in the general e-book piracy space, such as [32] and [33] echo findings of research on music and audiovisual piracy: displacement effects are mostly detrimental for best sellers, long tail content enjoys a discovery effect, and the individual propensity to pirate depends on individual norms and attitudes, peer pressure, price sensitivity and technical expertise. In general, however, only a very small segment of the population is involved in e-book piracy.

The high profile investigation and later suicide of Aaron Swartz, author of the Guerilla Open Access Manifesto [7], and the open rebellion of Alexandra Elbakyan [34], SciHub's administrator, brought the issue of scholarly piracy into the mainstream, resulting in a number of empirical studies on this phenomenon. The research was also aided by the openly accessible LibGen catalogue, and the dataset on SciHub usage released by Elbakyan in 2016 [35]. Cabanac [36] offers a rudimentary analysis of the LibGen Catalog, while Greshake does the same for the SciHub dataset [37]. Bodo [38] uses a download dataset of LibGen usage from 2012 and finds that the most popular titles in LibGen are widely available via Amazon in various print formats, suggesting that the library's main role is not the distribution of titles inaccessible via legal alternatives. The study, however, also found that cheap and easy electronic availability (both individual, and institutional) was limited in 2013–14, and downloaded works tended to be significantly more expensive that those which were not downloaded. The issue of e-book availability and the limited rights of libraries to lend e-books was also confirmed in a more recent study by Giblin at al. [39].

Himmelstein [30] analyzed the SciHub catalog, and found that in many scientific domains it offers more comprehensive access to paywalled articles that even the best US academic libraries. Muller and Iriarte [40] measured the availability and access of journal articles cited by University of Geneva researchers in 2015–16 via various sources including SciHub, and found that compared to legal availability, piratical access plays very little role. This is in line with a number of studies from multiple scientific disciplines, which found that the overall weight and impact of this piratical access channel remained marginal [41–43].

Regarding the geographic usage of shadow libraries, both Bodo [38] analyzing LibGen, and Bohannon [2] analyzing SciHub data, agree that these services are widely used in both developed and developing countries. This fact suggests the existence of multiple, separate logics that produce the use of scholarly piracy. In rich North American and Western European countries, users turn to SciHub and other similar venues most likely for convenience [11]. On the other hand, studies from developing countries suggest a substantial access problem in the Global South, which may drive scholarly piracy [17, 44, 45].

In this paper we use a large dataset of directly observed downloads from one of LibGen's mirror sites [46]. We use this dataset to model what kind of macroeconomic and institutional conditions may explain the use of shadow libraries. We are particularly interested in the potential function of shadow libraries to mitigate income-related access problems in the periphery. We test the following two hypotheses:

*H1*: *Globally, per capita shadow library usage is more prominent in lower-income countries, controlling for internet penetration.*

We also test the same hypothesis within the European Union, where a much richer dataset allows us to conduct analysis on significantly smaller, sub-national statistical units.

*H2*: *Within the European Union, the use of shadow libraries is more prominent in lower-income EU regions, controlling for the number of academics in the region.*

In addition, the richness of additional data sources in Europe allowed us to test if there are other, less intuitive spatial or social patterns that could offer more detailed insight into scholarly piracy. We compiled a rich dataset from various European official data sources, such as EUROSTAT and Eurobarometer, and used various modelling techniques, such as random forest simulation, to identify and test additional explanatory variables, which we could then integrate into our piracy models. All data and code are available in our public repositories for review and reuse. [46, 47]

## Data overview and descriptive statistics

Multiple sites offer access to books in the LibGen database. The dataset we analyze herein is a weblog of one such LibGen mirror site, which has been in continuous operation since at least 2012. The data was provided to us by an anonymous administrator through private correspondence during 2015. Each record in the dataset contained a timestamp, a unique document ID from the LibGen catalogue, and an IP address. We converted IP addresses to Geolocation data using Maxmind's GeoIP database [48], and discarded the IP addresses. After the removal of obvious bot traffic (such as repeated requests from the same IP address to the same title within a 24-hour time window), and traffic from known TOR exit servers, the logs contained 16133680 records over a period of 135 days from between 09/27/2014 and 03/01/2015. We aggregated this dataset by country for a global, country level analysis, and by NUTS2 statistical units within the European Union for the European analysis.

Our hypotheses address the following question: are pirate libraries used by individuals to compensate the structural limitations in the legal access alternatives? Based on earlier studies [20, 49, 50] and the accounts of pirate library operators themselves [1, 6] we assume that in lower income regions, access to knowledge faces multiple barriers: libraries and other knowledge institutions have less to spend on new acquisitions, while individuals may not find the prices of commercial alternatives affordable [50, 51]. Therefore, our independent variables try to capture this income effect both on the institutional and the individual level.

In line with the literature reviewed above, we compiled two sets of independent variables (one global and one European), to model the social, macro-economic environment which may impact pirate library usage. To normalize the download volumes, we used the World Bank database [52, 53] for data on population (SP.POP.TOTL). Institutional barriers to access, and low individual purchasing power has both been shown to fuel piracy [49, 50], so we use GDP (per capita, PPP, current international $—NY.GDP.PCAP.PP.CD) to capture both effects, assuming that lower GDP corresponds to lower institutional access budgets. We use fixed broadband subscriptions (IT.NET.BBND) to control for the fact that—apart from local copies distributed via hard drives—the LibGen collection is accessible online. Since it is a specialized collection of scholarly works, we use a number of candidate variables to capture the size of potential demand in research and higher education for this type of supply: Literacy rate (adult total, % of people ages 15 and above—SE.ADT.LITR.ZS), Research and development expenditure (% of GDP—GB.XPD.RSDV.GD.ZS), and School enrollment, tertiary (% gross—SE.TER.ENRR). To better differentiate between the effects of privately and publicly funded higher education, where the higher per-student costs in privately funded systems may push larger student populations to piracy, we used the OECD's Education at a Glance database [54] for data on government expenditure on tertiary education per student in constant 2014 PPP US. To measure the intensity of scholarly research activity Scimago Journal & Country Rank dataset [55] was used to get country level H indices.

For the European analysis, the Eurostat database [56] offered higher resolution datasets into the same dimensions. We use the Disposable income of private households by NUTS 2

regions (tgs00052) variable to capture potential individual income effects, and Gross domestic product (GDP) at current market prices by NUTS 2 regions (isoc_r_broad_h), Gross domestic product ({GDP}) at current market prices by NUTS 2 regions (isoc_r_broad_h) to capture aggregate, institutional income effects. We have better controls for online access through variables which also measure online proficiency: Households with access to the internet at home (isoc_r_iuse_i), Households with broadband access (isoc_r_gov_i), Individuals who accessed the internet away from home or work (tgs00002), Individuals who have never used a computer (isoc_r_blt12_i), Individuals who ordered goods or services over the internet for private use (tgs00026), Individuals who ordered goods or services over the internet for private use in the last year by NUTS 2 regions (edat_lfse_04), Individuals who used the internet for interaction with public authorities (demo_r_pjangroup), Individuals who used the internet, frequency of use and activities (isoc_r_iumd_i). We capture the size of the potential audience of scholarly pirate libraries via Population aged 25–64 by educational attainment level, sex and NUTS 2 regions (%) (isoc_r_iacc_h), Population on by age group, sex and NUTS 2 region (rd_p_persreg), Total R&D personnel and researchers by sectors of performance, sex and NUTS 2 regions (isoc_r_cux_i). To control for the different sizes of territories, we use Total and land area by NUTS 2 region (tgs00026). We also used the Eurobarometer 79.2 survey—ZA No. 5688 [57] for cultural access and participation variables, such as visiting a public library at least once a year, reading a book at least once a year, and not visiting public libraries more often because of perceived low-quality local supply. These latter two variables are to measure the potential substitution effect mentioned earlier in the literature review, but between piratical and physical libraries. We provide a detailed description of all data, including information on data preparation, missing variable handling, and codes in the S1 File.

## Download data overview

Fig 1 shows the daily number of downloads. Except for two periods with no data, the logs raise no apparent doubts about the validity of the information within.

In May 2015, at the end of the observed period, the LibGen database contained little more than 1.6 million records. The weblogs referred to the download of 760868 books from the LibGen catalogue. Compared to data from 3 years earlier from the same source [38], the catalogue grew by half a million records from ~836000 to ~1300000, while the average daily download volume grew more than threefold from ~41000 downloads per day to ~136000 downloads per day.

In Table 1 we listed the first 20 countries by absolute download volume. In the last two columns, we listed average daily download per million inhabitants and the rank of the country by per capita downloads.

Looking at the geographic location of downloads at Fig 2, one can observe that while most downloads cluster around large urban centers and locations that coincide with institutes of research and higher education, a substantial amount of activity originates from outside these intuitive download locations.

The content-wise analysis of downloaded works (not reported here) also supports the self-professed claims of LibGen that it is a predominantly scholarly library used to disseminate academic works indiscriminately across the globe to scholarly communities and individuals interested in learning.

## Global models

Our first efforts try to explain the global per capita download volumes by macroeconomic indicators, such as Population (Total), GDP per capita, PPP (current international $) and

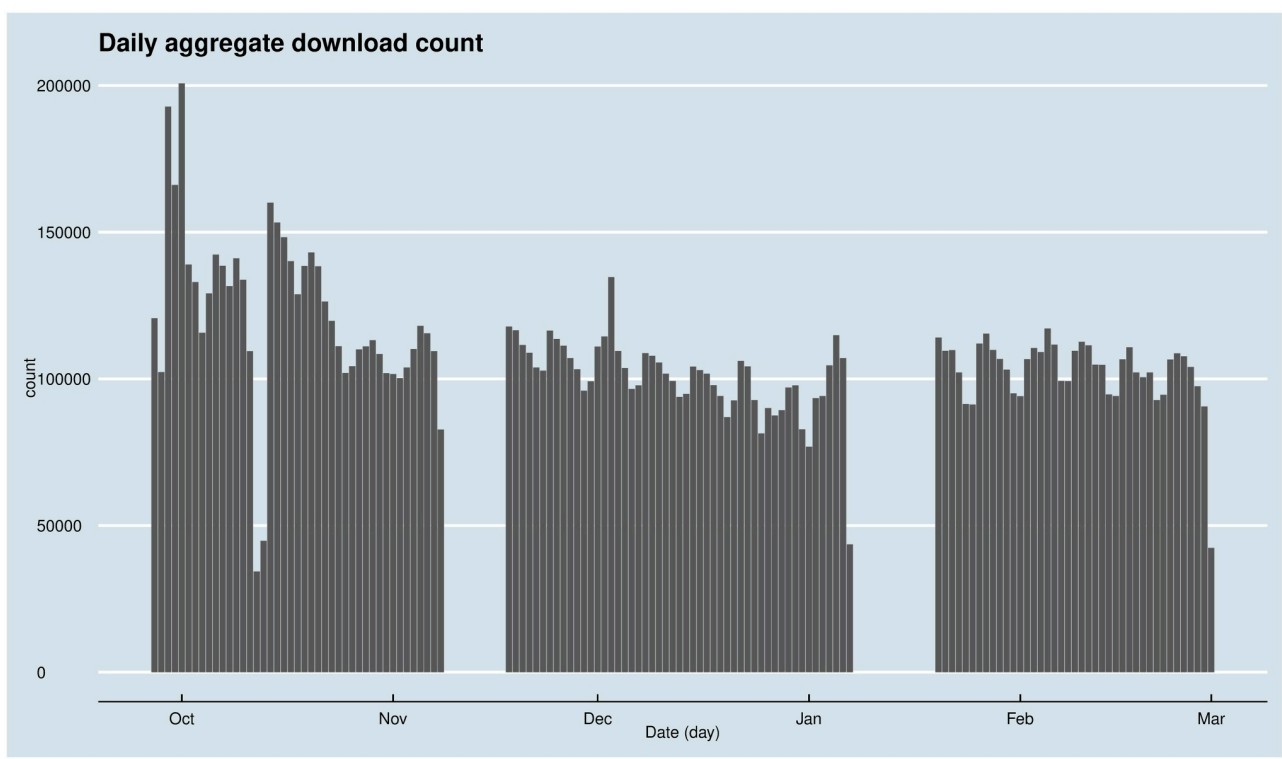

**Fig 1. Daily aggregate download volumes.**

**Table 1. Country level statistics for the first 20 countries by aggregate download volume.**

|  | Country name | Total downloads | download per DAY PER million | per capita download rank |
|---|---|---|---|---|
| 1 | United States | 1683353 | 39 | 49 |
| 2 | India | 1272124 | 7 | 131 |
| 3 | Germany | 765170 | 69 | 19 |
| 4 | United Kingdom | 594925 | 68 | 21 |
| 5 | China | 580808 | 3 | 158 |
| 6 | Iran, Islamic Republic of | 563798 | 53 | 35 |
| 7 | Italy | 469676 | 57 | 30 |
| 8 | Canada | 369962 | 77 | 17 |
| 9 | Indonesia | 341269 | 10 | 119 |
| 10 | Spain | 327326 | 52 | 37 |
| 11 | Turkey | 323204 | 30 | 63 |
| 12 | Brazil | 307376 | 11 | 112 |
| 13 | France | 290734 | 32 | 59 |
| 14 | Greece | 237657 | 163 | 3 |
| 15 | Mexico | 200792 | 12 | 108 |
| 16 | Australia | 200109 | 62 | 24 |
| 17 | Russian Federation | 196087 | 10 | 118 |
| 18 | Netherlands | 189747 | 83 | 14 |
| 19 | Vietnam | 179758 | 14 | 101 |
| 20 | Egypt | 169421 | 14 | 102 |

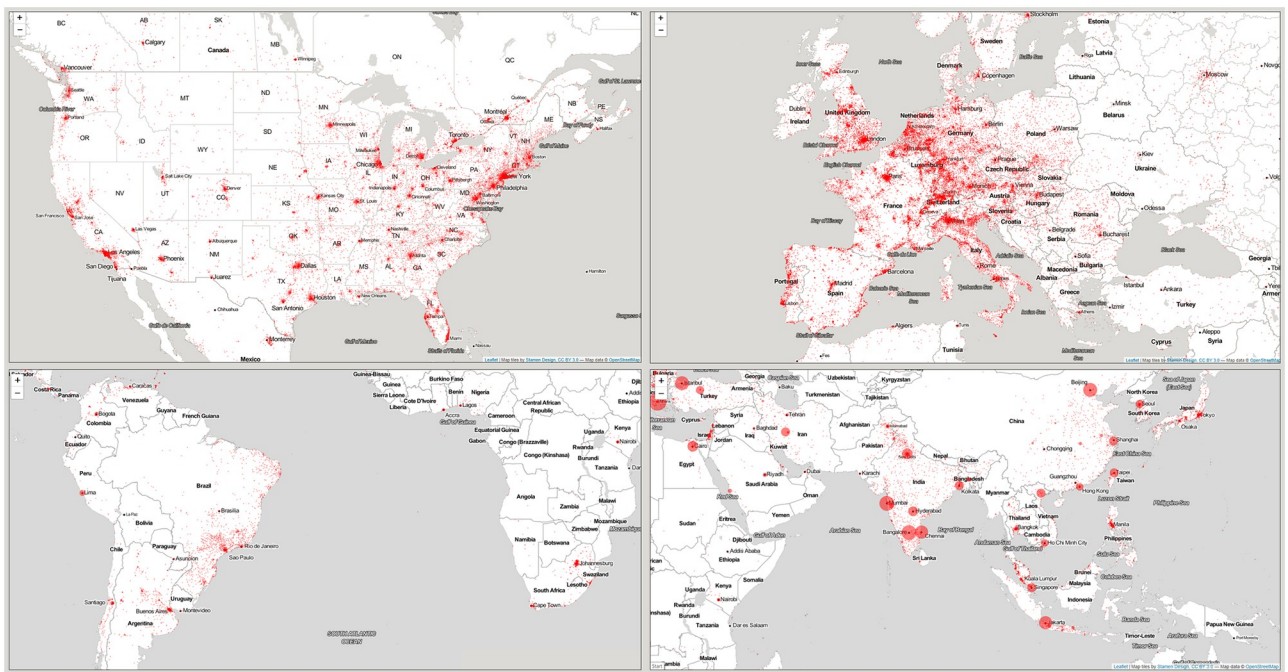

**Fig 2. Geographical distribution of download locations aggregated over the total observation period.**

internet penetration (Fixed broadband Internet subscribers). We then try to add variables related to education and research, such as Literacy rate (adult total, % of people ages 15 and above), and School enrollment, tertiary (% gross), Research and development expenditure (% of GDP), Government expenditure per student, tertiary (% of GDP per capita), scholarly research impact, as measured by the aggregate h-index of the country, macro-statistics from the World Bank, and OCDB statistical databases. Descriptive statistics of these variables are in S1 Table.

If we plot the number of downloads per population per country (colored by continent), we see that there is substantial variation among countries (Fig 3a), and between countries of different continents (Fig 3b).

In the first model, we use the following specification:

$$y = \alpha + \beta_{gdp} * GDP + \beta_{pop} * Population + \beta_{internet} * InternetPenetration + \epsilon \qquad (1)$$

We tested this model both as a linear model and using a Poisson regression. As the data consists of count data, both Poisson or binomial distributional families could be used for modeling. We observe that the two distributions yield similar predictive performance (overinflated Poisson leads to better), however following Gelman and Hill [58], we find that Poisson distribution fits our data generating process better because downloads are not based on independent trials, and interpreting them as a number of successes—as in a negative binomial approach—can be tricky. For this reason we omitted a negative binomial approach.

The outputs of the model can be seen in column (1) of Table 2. Standard errors are in parentheses. Models 2 and 3 use the rounded value of download per capita as DV.

In the general linear model (Model 1), only the Internet Penetration and the GDP have significant effects (the latter only at a 90% level), both being positive. While there is a 0.75

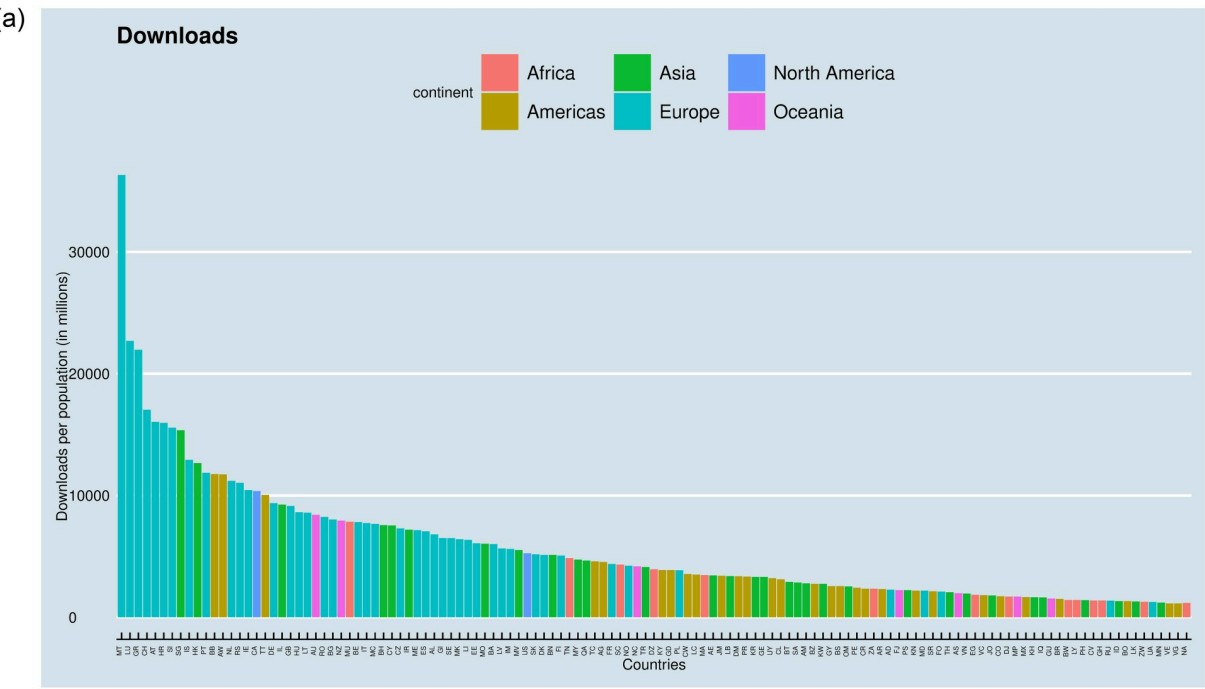

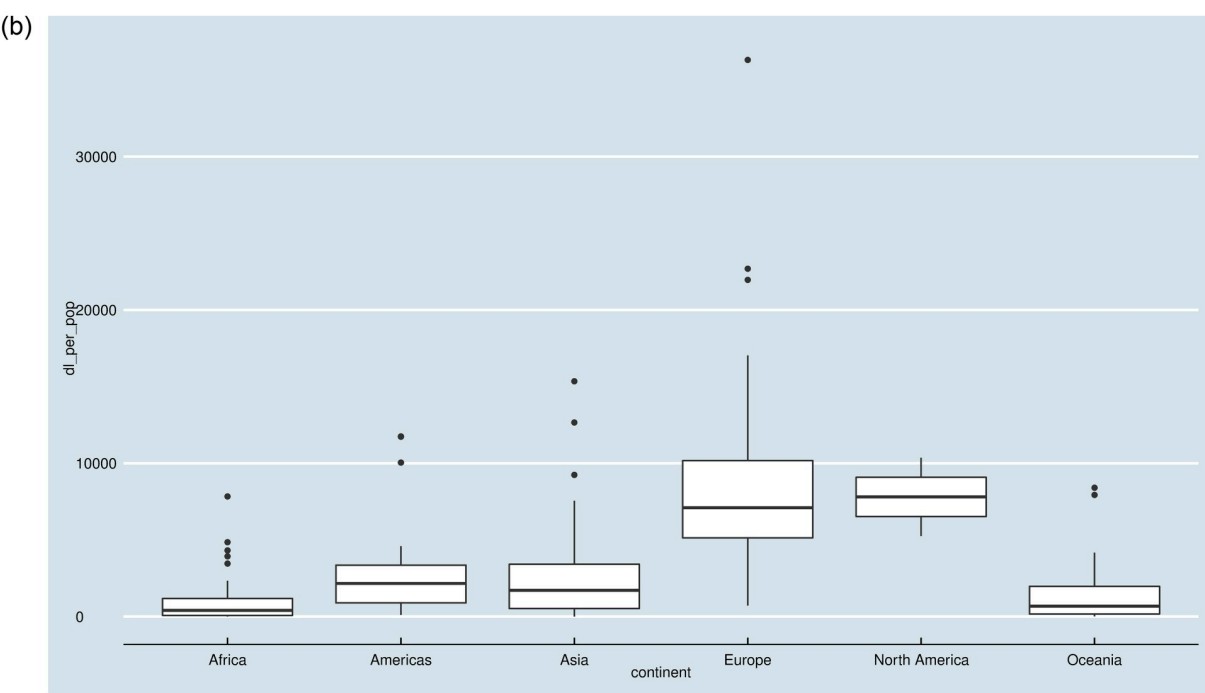

**Fig 3.** a, b. Country-level and regional variance of the dependent variable.

correlation between GDP and Internet penetration, the VIF values of the model show that the model does not suffer from multicollinearity.

With the Poisson regression in column (2) of Table 2, all the variables are highly significant. The VIF values are less than two, so multicollinearity in not a concern. The signs of the

**Table 2. Global models I.** (DV: download per capita).

|  | Model 1 | Model 2 | Model 3 |
|---|---|---|---|
| (Intercept) | -5.26e+03 | 2.5 *** | 2.5 ** |
|  | (3.22e+03) | (0.0188) | (0.893) |
| log(population per million) | -83.1 | -0.0181 *** | -0.0181 |
|  | (125) | (0.000537) | (0.0255) |
| log(gdp) | 712 | 0.531 *** | 0.531 *** |
|  | (376) | (0.00205) | (0.0972) |
| broadband_subscribers | 1.95e+04 *** | 2.82 *** | 2.82 *** |
|  | (3.36e+03) | (0.0134) | (0.635) |
| N | 190 | 190 | 190 |
| Null deviance | 4.51e+09 | 9.26e+05 | 9.26e+05 |
| res.deviance | 2.48e+09 | 3.86e+05 | 3.86e+05 |

*** $p < 0.001$;

** $p < 0.01$;

* $p < 0.05$.

coefficients are the same as with the linear model: countries with higher gross income and better internet access download more. Population enters as a highly significant explanatory variable with a negative sign, which may be the result of two factors. On the one hand, the knowledge demand of populous countries like China, India, and Indonesia are not best served by a predominantly English language shadow library. On the other hand, it is possible that the share of the population working in knowledge-intensive domains of society does not scale linearly with population.

One possible downside of the Poisson regression is that it cannot deal with overdispersion (only one parameter is estimated). This can lead to underestimated standard errors, which we tested with a Wald test. Since the scale factor in the Poisson model is much higher than 1 (residual deviance / df = 385842 /186), we corrected for overdispersion by using a QuasiPoisson regression model, presented in column (3) of Table 2. In this last model, GDP and internet penetration are highly significant, and have positive effects.

Taken together, these models suggest a result which contradicts our hypothesis that low(er) income countries may use shadow libraries more to compensate for infrastructural, and funding limitations.

To explore further, we added a number of macroeconomic variables related to tertiary education and research activities. We queried gross tertiary education enrollment ratio, the expenditure on tertiary education per student and the percentage of GDP spending on R&D from 2015 from the World Bank Open Data dataset. We also included the H index of countries from 2015. Due to missing data, the sample size was reduced from 190 to 86. For this model and all the following ones, we only include the results of the QuasiPoisson regression as this is the best fit for our data. The results are summarized in Table 3.

Including these extra variables in the original model (Table 3 Model 4) did not produce significant new insights, as the GDP variable seem to capture much of the effect of higher education and research investment. In Model 5 we excluded the GDP and internet penetration variables. In the resulting weaker model, the share of the population with tertiary education, and the h-index variable became significant (the latter only at a 90% level), with intuitive results: a larger share of highly educated people, and more relevant scientific output results in higher shadow library use. Interestingly, the share of active tertiary education students, a

**Table 3. Global models II.** (DV: download per capita).

| | Model 4 | Model 5 | Model 6 |
|---|---|---|---|
| (Intercept) | 2.44 | 8.15 *** | 7.12 *** |
| | (1.67) | (0.267) | (0.358) |
| log(population per million) | -0.0926 | -0.326 *** | -0.341 *** |
| | (0.0875) | (0.0719) | (0.0672) |
| log(gdp) | 0.583 ** | | |
| | (0.181) | | |
| Broadband_subscribers | 1.61 | | |
| | (1.43) | | |
| Tertiary_education_enrollment_ratio | 0.0023 | 0.0102 * | 0.0297 *** |
| | (0.0043) | (0.00444) | (0.0061) |
| Expenditure_tertiary_education_per_student | -1.37e-05 | 9.11e-06 | -2.49e-06 |
| | (1.57e-05) | (1.78e-05) | (1.56e-05) |
| Percentage_of_GDP_spending_on_R&D | 0.0875 | 0.148 | 1.56 *** |
| | (0.103) | (0.11) | (0.323) |
| H_index | -0.000156 | 0.000743 | 0.000896 * |
| | (0.000491) | (0.000446) | (0.000447) |
| Tertiary_education_enrollment_ratio: Percentage_of_GDP_spending_on_R&D | | | -0.0211 *** |
| | | | (0.00476) |
| N | 86 | 86 | 86 |
| Null deviance | 4.7e+05 | 4.7e+05 | 4.7e+05 |
| res.deviance | 1.97e+05 | 2.53e+05 | 1.91e+05 |

*** $p < 0.001$;

** $p < 0.01$;

* $p < 0.05$.

potential source of shadow library traffic, was not significant. In the last model we introduced a new interaction term between the share of population with tertiary education and RD expenditure, because these two activities are two most obvious sources of shadow library use, but they may be independent from each other. In Model 6 both variables turn highly significant, suggesting that both the size of the highly educated population and the RD activities contribute positively to shadow library demand, albeit at a diminishing rate, as the interaction term has a negative sign. We found no effect of public spending on tertiary students, which could have differentiated between countries with publicly and privately funded higher education systems. These models refine the effect of GDP and identify research and tertiary education as major drivers of shadow library use.

Lastly, we explored the effect of regional differences, because the descriptive statistics suggest (see Fig 3a and 3b) that there are substantial regional differences. Our last global model is a varying intercept and slope (random effects) model. The model contains the intercept, and the GDP varying with the continent, while the effect of the population and internet penetration is fixed. The DV is download per capita.

Table 4 shows the following coefficients.

From the table above, the varying intercept points to higher European and North American download baselines. More interesting is the huge difference in the effect of GDP. The impact of gross income on downloading is much higher for countries in the African continent than, for example, in Europe.

**Table 4. Global models III.** Random effects model by continent (DV: download per capita).

| | log(1+ gdp_scaled) | (Intercept) | Population_per_million (scaled) | broadband_subscribers (scaled) |
|---|---|---|---|---|
| Africa | 1.1025678 | 8.248046 | -0.3200811 | 0.2780263 |
| Americas | 0.2314024 | 7.858342 | -0.3200811 | 0.2780263 |
| Asia | 0.5302122 | 7.981479 | -0.3200811 | 0.2780263 |
| Europe | 0.1876811 | 8.645233 | -0.3200811 | 0.2780263 |
| North_America | 0.3898888 | 8.455814 | -0.3200811 | 0.2780263 |
| Oceania | 1.2764129 | 7.672324 | -0.3200811 | 0.2780263 |

It is possible that the cause of these differences is geographical in nature, because, for example, shadow library related practices propagate via physical proximity and the close trust relationships of individuals. While this may be the case, it is hard to test that hypothesis with current data. On the other hand, geographic location may also be a proxy for the level of development and in that case, we can conduct a similar analysis using the World Bank's income categories instead of geographic location, as well as test for the effect of GDP, R&D and educated population in different income categories.

Tables 5 and 6 show the outcome of these models.

Both models point to interesting findings. The effect of GDP is very different in the four income categories. In low-income countries, increasing GDP causes much larger shadow library use than in high-income countries. The model in Table 6 suggests that in low-income countries extra investment in tertiary education and R&D activities generates relatively larger shadow library usage than similar investment in high-income countries. In the latter group, extra investment into R&D and tertiary education is associated with relatively lower download volumes, while in low-income countries the effect is exactly the opposite: higher investment into knowledge-intensive social activities generates more demand for black market knowledge. The reason for that is straightforward. In high GDP countries, extra money spent on knowledge-intensive activities is more likely to include spending on infrastructures of legal access, lessening the need for grey market venues. On the other hand, in low-income countries where infrastructure is probably the most lacking, any step to increase knowledge-intensive domains and knowledge-hungry populations is likely to hit infrastructural constraints, leaving some of the demand at the mercy of access provided by shadow libraries.

At first sight, the global models did not support our first hypothesis: that countries with fewer resources to spend on research and higher education would be more intense users of shadow libraries to offset their infrastructural limitations. On the contrary, our early findings suggested that as countries' GDP per capita, tertiary enrollment, or research expenditure grows, they also make use of shadow libraries more intensely. At the aggregation level of individual countries this is hardly surprising: access to knowledge is only one element in the complex infrastructural mix which then produces demand for the knowledge shadow libraries may offer.

We arrive at more nuanced conditions when we start to disaggregate the impact of gross income through finding various proxies of general development. The intuition behind this approach is that investment into knowledge-intensive societal domains, such as R&D and higher education, serves different purposes and has different effects at different stages of development. In low-income countries, higher investment may lead to fast-growing knowledge absorption capacity, which may not be met with appropriate infrastructural support. This means that low-income countries generate less shadow library usage in general, but within that group, larger investment into knowledge-intensive activities has greater positive impact

**Table 5. Global models IV.** GDP random effects model by income category (DV: download per capita rounded, quasipoission).

| | log(1+gdp_scaled) | (Intercept) | Population per million (scaled) | Broadband subscribers (scaled) |
|---|---|---|---|---|
| High_income | -0.003871827 | 0.30688303 | -0.1200297 | 0.3071157 |
| Upper_middle_income | 0.105214527 | -0.06604125 | -0.1200297 | 0.3071157 |
| Lower_middle_income | 0.397072298 | -0.07077302 | -0.1200297 | 0.3071157 |
| Low_income | 0.348602516 | -0.17360155 | -0.1200297 | 0.3071157 |

on usage. In high-income countries the logic is the opposite. While they have larger per capita demand, larger investment in knowledge-intensive activities does not increase black-market demand further. On the contrary, since extra investment most probably creates better infra-structural conditions, rather than extra knowledge absorption capacity, larger investment leads to relatively lower black-market demand.

In summary, having access to a virtually unconstrained source of knowledge does not mean automatically receiving all potential benefits. The impact of higher income manifests itself in two forms. On the one hand, it creates knowledge demand through the prominence of knowl-edge-intensive institutions, and knowledge-demanding social strata. While higher income cer-tainly expands the knowledge absorption capacity of countries, it may not establish the adequate institutional frameworks to service that demand at the same pace. The infrastructure of legal access may be lagging behind the growth of this demand, which creates ideal condi-tions for shadow library use. It seems that only at higher-income levels does the extra invest-ment in knowledge-intensive domains create the adequate access infrastructures which ultimately moderate shadow library use.

In its original form, our hypothesis is only supported among high-income countries. But the larger impact of R&D and educational investment on downloads in low-income countries also lends support to this hypothesis, albeit is a slightly different form. In low-income coun-tries, the per capita download starts at a lower base, because the knowledge absorption capacity of these countries is limited. However, any extra income produces growth in that absorption capacity, which in turn creates comparatively larger demand for shadow libraries.

At this stage, we should point to some of the limitations in our data that may affect these findings. First, data may be skewed by the use of VPNs by users whose ISP blocks access to Lib-Gen. Second, in countries with low bandwidth, local copies of shadow libraries may serve much of the demand, therefore the download figures underestimate the actual use of this resource.

## European models

In this section we focus only on downloads from within the European Union (Fig 4). This allows us to address many of the limitations of global models. First, we can zoom in to regional

**Table 6. Global models V.** R&D and education random effects model by income category (DV: download per capita rounded, quasipoission).

| | Tertiary education enrollment ratio (scaled) | Percentage of GDP Spending on R&D (scaled) | (Intercept) | Population per million (scaled) | Broadband subscribers (scaled) |
|---|---|---|---|---|---|
| High income | 0.010932241 | -0.05358038 | 8.543931 | -0.2996871 | 0.4264995 |
| Upper middle income | -0.001022486 | 0.35217880 | 8.230012 | -0.2996871 | 0.4264995 |
| Lower middle income | 0.050455057 | 1.63851738 | 8.360397 | -0.2996871 | 0.4264995 |
| Low income | 2.797771598 | 1.45719062 | 9.857220 | -0.2996871 | 0.4264995 |

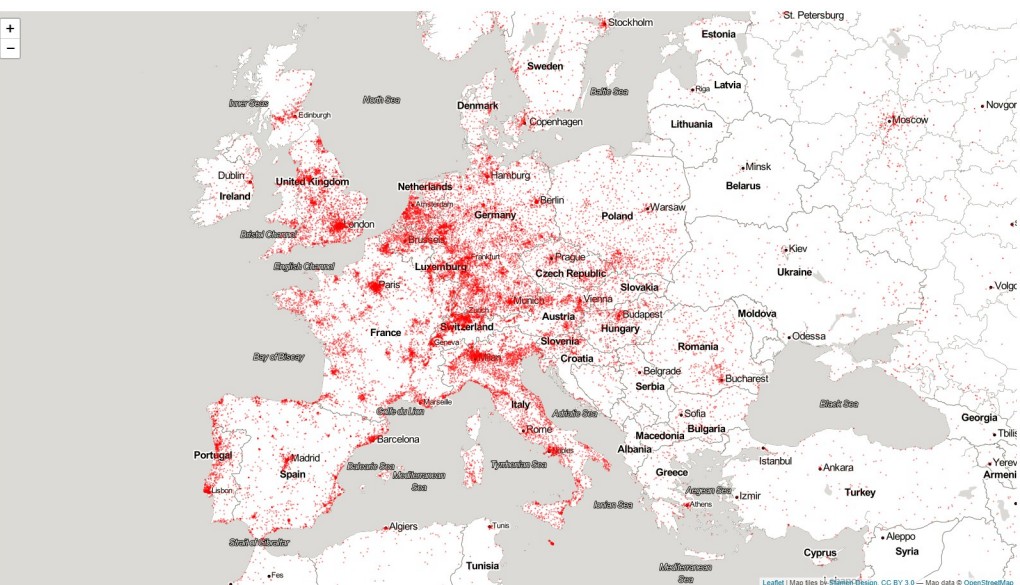

**Fig 4. European download locations.**

levels. Geolocation tends to work better in Europe, so we can be more confident of the geographic location of a particular download and focus on sub-national socio-economic units. Second, due to a number of datasets produced by the European statistical agency Eurostat, we have a much better selection of institutional, economic, and attitudinal variables. Third, as the authors are from the European Union, we have the advantage of being familiar with the home field to interpret results.

The European Union is the world's largest harmonized statistical data collection area with four levels of statistical aggregation starting from national (NUTS0) level to very small territorial units, down to NUTS3 level. We selected the NUTS2 regions of the European Union for our environmental analysis. The NUTS2 regions were created for socio-economic statistical purposes and they are designed to maximize intra-unit homogeneity. While NUTS2 social and economic data is not always complete, partly because NUTS2 boundaries change relatively frequently, we can usually work with 140–260 territories. The Eurobarometer, or the European Social Survey, is designed to represent NUTS0 (country) levels, but data can be re-aggregated at NUTS2 levels with relatively little bias. The NUTS2 level is a good compromise between NUTS0 (country) and the much smaller NUTS3 levels. While NUTS3 levels allow the comparison of roughly a thousand environments, there is far less data available on NUTS3 level. Furthermore, on NUTS3 level, we need to tackle problems of non-normal distribution, as on the NUTS3 level our data becomes asymmetric. Therefore, we aggregated the download data over the NUTS2 boundaries, joined them with environmental data, input missing variables, and normalized the data. These processes are described in more detail in the supplementary material on methods in S1 File.

The richness of the European dataset allowed us to purse a deductive modelling strategy, and additionally use advanced statistical methods to explore new patterns in the data.

### Hypothesis testing

First, we tested our original hypothesis on the European regional data, namely:

*H2*: *Within the European Union, the use of shadow libraries is more prominent in lower-income*
    *EU regions, controlling for the number of academics in those regions.*

Historical accounts that reconstruct the development and raison d'être of shadow libraries
[1, 9, 59, 60] suggest that inadequate legal access alternatives may be the main drivers of digital
piracy in this region in general. The authors, who originally had close relationship with acade-
mia in the region, also have extensive personal experience with the lack of infrastructural con-
ditions of scholarly work, and the consequent extensive use of piratical resources—both to
provide competitive higher education degrees for students with an eye on the European job
market, and to produce research relevant in the European and global arena. Our first-hand
experience matches other accounts on the economic and academic periphery: shadow libraries
may offer a way to overcome income-related infrastructural limitations for scholars. On the
other hand, various studies on the relationship of R&D activity, and economic development
found that lower economic development sets an upper limit to the effective utilization of pub-
lic and private R&D investments, because of the economic sector's limited knowledge absorp-
tion capacity [61].

We chose download per capita as a dependent variable for our model, with the following
specification:

$$y = \alpha + \beta_{gdp\_pps} * log\left(GDP_{PPS}\right) + \beta_{researcher} * Researcher\_employment\_pct + \beta_{internet_{banking}} * Internet\_banking\_pct + \epsilon \text{ (2)}$$

Where *GDP_PPS* is the price-adjusted version of the GDP indicator, using purchasing
power standards rather than Euros to account for the differences is purchasing power (used in
a logarithmical form); Researcher_employment_pct is the percentage of R&D personnel and
researchers in the workforce, and Internet_banking_pct is the percentage of the population
that used the internet for online banking. We treat this latter variable as a rough proxy for
internet proficiency.

This model is somewhat comparable to the global model, as it refers to the same underlying
dynamics, albeit with variables that better approximate the factors in question. Instead of R&D
expenditure of the global model (which was found to be insignificant in all models), we use the
share of researchers in the local workforce, and instead of using internet penetration, we use
data on the advanced use of the internet.

As before, we use a QuasiPoisson regression model to correct for overdispersion, and
account for the fact that we model count data. The VIF values of the regression are all less than
two, so multicollinearity is not of concern.

In Table 7, Model 7, which is our base model, all independent variables are highly signifi-
cant and we explain ~72% of the variance ($R^2$ = 0.7255838). The per capita downloads grow
with GDP, as well as with the share of researchers in the workforce. On the other hand, shadow
library usage is moderated by internet proficiency.

The interpretation of the former two effects is straightforward and in line with the findings
of our global models. Shadow library usage is positively correlated with income. It is also intui-
tive that the researcher population drives shadow library demand. The negative sign of internet
proficiency variable, however, demands some explanation. That variable can be a proxy of
many different online skills: better knowledge of digital piracy, including the use of shadow
libraries; skills to use the internet for online purchases; and skills for hiding the traces of illicit
activities, via the use of Virtual Private Networks, and TOR browsing. VPNs and the TOR net-
work allow users to preserve their online privacy by routing their online traffic through a num-
ber of intermediary computers to a random exit point on the internet. This has the implication

**Table 7. European models I.** (DV: download per capita).

| | Model 7 | Model 8 | Model 9 | Model '10 | Model 11 |
|---|---|---|---|---|---|
| (Intercept) | 6.438 *** | 6.295 *** | -0.143 | 6.353 *** | 4.050 *** |
| | (0.794) | (0.838) | (2.457) | (0.802) | (1.110) |
| log(GDP purchasing power parity) | 0.247 ** | 0.242 ** | 0.175 * | 0.258 ** | 0.490 *** |
| | (0.077) | (0.081) | (0.075) | (0.078) | (0.105) |
| % of R&D personnel and researchers in the workforce | 0.697 *** | 0.683 *** | 0.570 *** | 0.702 *** | |
| | (0.057) | (0.063) | (0.068) | (0.055) | |
| % of the population that used the internet for online banking | -0.011 *** | | -0.015 *** | -0.009 ** | -0.003 |
| | (0.003) | | (0.003) | (0.003) | (0.005) |
| % of the population that used the internet for online shopping | | -0.006 | | | |
| | | (0.003) | | | |
| log(disposable income) | | | 0.792 ** | | |
| | | | (0.280) | | |
| R&D expenditure | | | | -0.070 | 0.059 |
| | | | | (0.053) | (0.076) |
| null.deviance | 2990524.371 | 2990524.371 | 2990524.371 | 2990524.371 | 2990524.371 |
| deviance | 1415805.433 | 1507337.393 | 1343129.996 | 1396838.896 | 2455507.137 |

*** $p < 0.001$;

** $p < 0.01$;

* $p < 0.05$.

that traffic from such services associated with a particular IP address and geographic location usually originates elsewhere.

To further explore what the online banking variable may refer to, we replaced it with the percentage of population that used the internet for online shopping in Model 8. The intuitive assumption here was that a negative relationship (a replacement effect) exists between online shopping and digital piracy. Though the sign of the variable was negative, the relationship was not significant. Since online shopping and online banking variables are highly correlated (Pearsons's: 0.86, $p < .001$), we can assume that online banking already captures some of that effect.

In Model 9, we tested further variables, such as the effect of disposable income, and the share of population with tertiary education. The effect of disposable income is positive and significant at a 99% level, while the effect of education is nonsignificant. While this model suffers from higher multicollinearity, it is clear that the individual income effect and the macro-income indicator both point to the same direction: people download more from more affluent regions.

In Models 10 and 11 we introduced the R&D variable but found no statistically significant effect.

That being said, the R&D expenditure becomes significant, and with a negative sign, if the dependent variable is downloads per researcher (see Table 8, Model 12). While the effect signs for the other relevant variables (GDP_PPS, internet proficiency) remain the same, when we normalize downloads for the number of researchers, a higher R&D spending has a moderating effect on the per researcher downloads. This may be the first sign that points to a structural link between the amount of investment into knowledge infrastructures and scholarly piracy. We should note that the download per researcher models have a worse $R^2$ than the per capita models.

**Table 8. European models II.** (DV: download per researcher).

| | Model 12 | Model 13 | Model 14 |
|---|---|---|---|
| (Intercept) | 5.530* | 8.034** | 6.486*** |
| | (2.240) | (2.546) | (1.147) |
| log(GDP purchasing power parity) | 0.161* | 0.174* | 0.175 |
| | (0.071) | (0.078) | (0.113) |
| log(disposable income) | 0.148 | -0.143 | |
| | (0.255) | (0.291) | |
| educational attainment | 0.008 | -0.000 | |
| | (0.008) | (0.009) | |
| R&D expenditure | -0.253** | -0.310*** | -0.155 |
| | (0.079) | (0.088) | (0.901) |
| % of the population that used the internet for online banking | -0.018*** | | |
| | (0.004) | | |
| % of the population that used the internet for online shopping | | -0.006 | |
| | | (0.004) | |
| log(GDP purchasing power parity): R&D expenditure | | | -0.024 |
| | | | (0.084) |
| null.deviance | 495798.787 | 495798.787 | 495798.787 |
| deviance | 362061.621 | 398536.668 | 414631.765 |

*** p < 0.001;

** p < 0.01;

* p < 0.05.

In models where the dependent variable is the raw download count (see S2 Table), we find results consistent with those above: wealth and researcher population have significant positive effects, internet proficiency has significant negative effects, R&D spending, educational attainment, disposable income, or online shopping variables are not or only weakly (at 95% level) significant.

So far, we have established that income and the researcher population are the most significant positive drivers of shadow library usage in Europe. In the next step, we build a simple model in which these two variables interact. In this model (see Table 9), raw, not normalized download count is the dependent variable, while GDP purchasing power parity variable is used is its natural form.

In Model 15, all coefficients are highly significant, with a negative interaction term. This suggests that within the EU, even if two regions have similar researcher density, high-income regions use shadow libraries more. The difference between low- and high-income regions is significant and diminishes only slightly with the growth of income (Fig 5).

## Discussion

The European models are in line with our global models and suggest that similar logics are at play within the European Union, as well as globally. We identified two main drivers of the demand for pirated knowledge: the presence of knowledge-intensive economic activity and GDP. Just as in the case of global models, the number of researchers sets the baseline demand: the production of knowledge requires knowledge. However, income-related infrastructural limitations do not translate into relatively higher shadow library use because income also defines knowledge absorption capacity. This finding is in line with earlier empirical studies on

**Table 9. European models III.** (DV: download count).

| | Model 15 |
|---|---|
| (Intercept) | 8.546 *** |
| | (0.159) |
| GDP purchasing power parity | 1.050631e-05*** |
| | (1.278794e-06) |
| % of R&D personnel and researchers in the workforce | 0.918*** |
| | (0.116) |
| GDP purchasing power parity: % of R&D personnel and researchers in the workforce | -3.479490e-06*** |
| | (7.194871e-07) |
| null.deviance | 7192467.382 |
| deviance | 3556373.970 |

*** p < 0.001;
** p < 0.01;
* p < 0.05.

R&D activity and economic development [61]. We found some support for this in the download per researcher model, where we found a strong and significant negative effect of R&D investment on per researcher download volumes. In the interaction model, we have also seen that some of the extra income probably sustains infrastructures that better cater for the extra demand.

Researchers in low-income regions may face many problems, legal access being only one of them. The authors have personal experience of at least some of the hurdles that may limit an intensive use of openly accessible knowledge wealth. Researchers at the economic and academic periphery may not be able to fully sustain themselves by having one single academic research job. The need to hold second and third jobs to sustain themselves financially may limit the time they can dedicate to library use, piratical or otherwise. Also, many of them find a predominantly English language shadow library less useful when their educational and research activities are not intended for the English-speaking global market.

## Inductive models

The variable selection in the European models was initially following the same conceptual guidelines as the global model. The dataset, however, allows us to switch modelling approaches and look for patterns in the data on which new hypotheses can be formulated.

Firstly, we have tested for spatial autocorrelation (Moran's test), where we found a weak spatial autocorrelation on a relatively low significance level. Second, we applied a more systematic variable selection method, by creating a random forest of regression models. Third, we used a 'brute force' approach to run all possible linear regressions and multiple regressions on all available data, to see if there is any spurious or real connection among the variables. Since this last approach did not reveal any new connection, we do not report the results, but the code executing this exploratory analysis is in the code repository [47].

## Spatial autocorrelation

The analysis of spatial autocorrelation reveals if shadow library usage is geographically clustered, for example, because underlying socioconomic activities are also clustered [62], or because user communities are clustered (for linguistic reasons, or because the knowledge

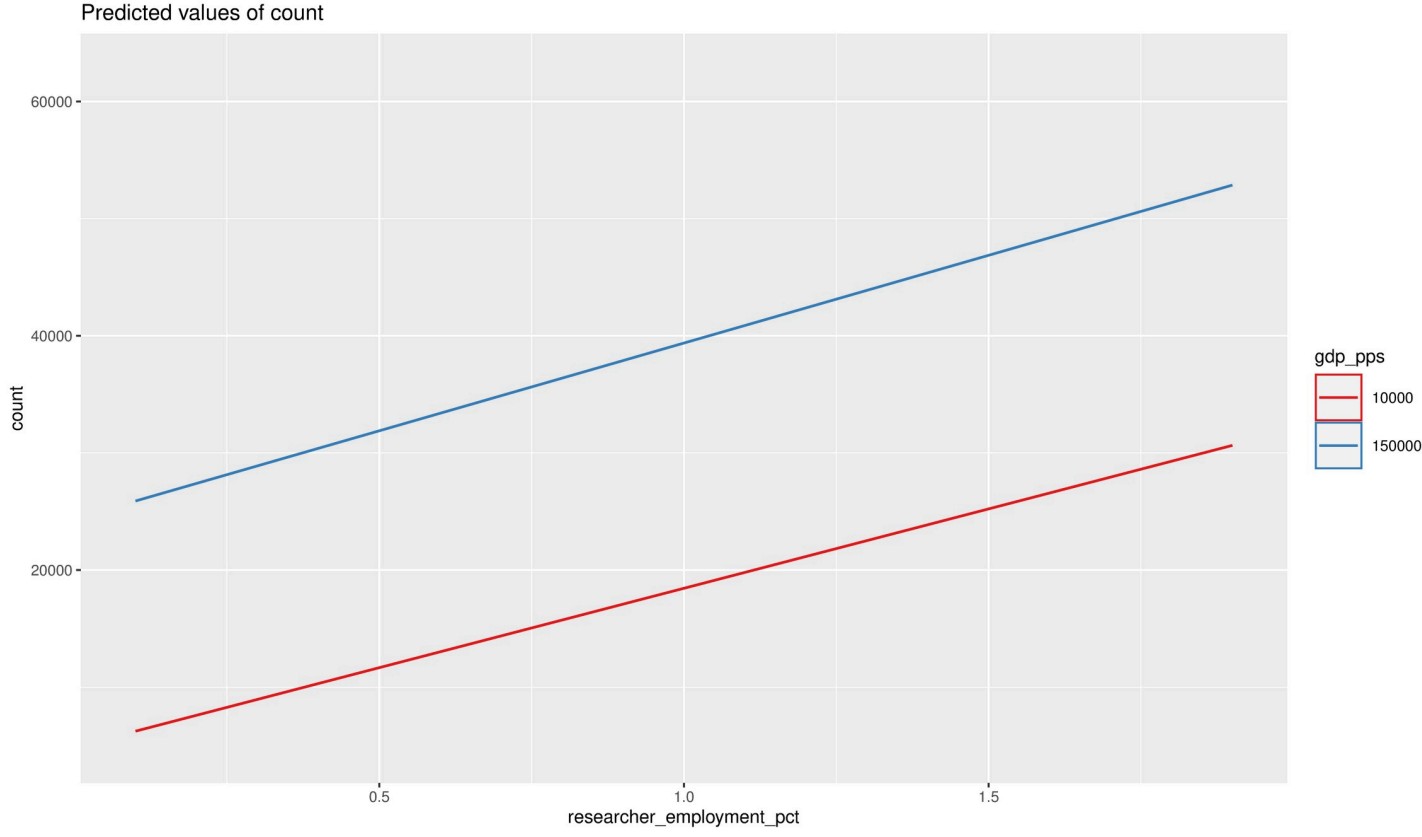

**Fig 5. Interaction effects between GDP_PPS and researcher employment percentage (DV: Download count).**

about shadow libraries dissipates in close-knit trust networks). The ability to examine spatial autocorrelation is an important check on the robustness of our methodology. Given that we do not have access to individual download data, only territorial aggregates of downloads, we want to ensure that downloading in geographical space is not happening randomly. We have examined the spatial autocorrelation using the 'spdep'package [63].

In the case of the download count variable, Moran's I statistic takes the value of 0.042 with a p-value of 0.094, so we can only reject the randomness of downloads at a 90% significance level. The positive z value means that the downloads are clustered, i.e. NUTS2 regions with high download numbers tend to be neighbors of NUTS2 regions with high download numbers. If we run the same test on the GDP, adjusted by purchasing power, we see a very similar level of spatial autocorrelation: Moran's I statistic is 0.044017, p-value = 0.077.

The results, at least on the NUTS2 level, do not point to well-defined download centers within Europe. Their strong similarity with how GDP is geographically distributed suggests that it is unlikely that downloads follow a random pattern and are closely related to the socio-economic factors that define the wealth of a region in general.

## Random forest models

The random forest method (RF) was mainly developed to solve classification or regression problems, but it has been long recommended for use in variable (pre-)selection [64]. While machine learning models may have high predictive power, they are often 'black boxes' that make conceptual explanations difficult, and their results are not directly comparable with the

global model. For these reasons, we did not use RF to choose a model candidate from the random forest regressions to test our hypotheses, but rather, used this approach to identify which variables out of the 50 available may play an unexpected, but important role in explaining downloads.

In a series of models unreported here, we first narrowed down the basic geographical and demographic forces attracting higher download counts, such as the land area of the NUTS2 region, population and population density, or researcher population density of the regions. We also normalized count with land area, population, and researcher count to gain deeper insight into the non-trivial social factors that attract heavier reliance on the research black market [58].

In the second round, we ran the random forest algorithms on the various forms of the count data to identify the most important social, cultural, and economic variables. We used the random Forest R package [65] for this purpose. First, we established the optimal parameters for starting the algorithm with the tuneRF function. We used all predictors to build a forest of regression trees. We used the Interpretable Machine Learning method and package [66] to interpret the importance of each feature (see particularly Chapter 5.5. of [64]).

The form of random forest method we used was created by using random samples of our dataset, and fitting regression trees on these subsets of the dataset. By repeatedly splitting the dataset and testing a limited number of features at a time, the random forest algorithm does not usually require strict conditions on residual errors and is insensitive to multicollinearity. One draw-back of the random forest method, like many machine learning models, is that it uses its own metrics of accuracy. For comparability, we used a model-agnostic feature importance metric by Molnar et. al. [64], a metric that results in comparable metrics for random forests and any other statistical model. This feature importance algorithm shuffles the values of the predictors and measures the change in a loss function (in our case, mean average error increase in the targeted dependent variables) for each shuffle—the larger the increase in mean average error, the more important to use the (correct values) of the predictor.

In addition to the random forest approach, which comprehensively compared the predictive power of variables groups, our 'brute force' approach has also measured all possible regression models. Given that both methods are comprehensive and did not reveal any further research directions, we found our initial theoretical framework validated, and based on this variable selection we created comparable regional models to the global model.

The random forest approach did not reveal any new connections on the dataset only containing the EUROSTAT variables. However, it identified a number of significant variables in the smaller, but richer dataset, which also includes the library-use related EUROBAROMETER variables.

The feature importance graph (Fig 6) identifies as relevant the same variables we already included in our linear models: the share of researchers in the workforce, GPD per capita in purchasing parity units, and R&D investment. In addition, the share of library using and book reading populations from among the EUROBAROMATER variables are also somewhat relevant.

Subsequently, we have included the newly identified EUROBAROMETER variables into the QuasiPoission regression models, with the per capita, per researcher and raw count as dependent variables. Table 10 contains the results of these models.

In various model configurations, most of the EUROBAROMETER variables remained insignificant. That being said, the variable on the share of population that reported to visiting a library in the previous 12 months was highly significant in both the download per capita and the download per researcher models, with a negative sign. Higher library use in the population goes hand-in-hand with lower pirate library usage. In addition, in the download per researcher

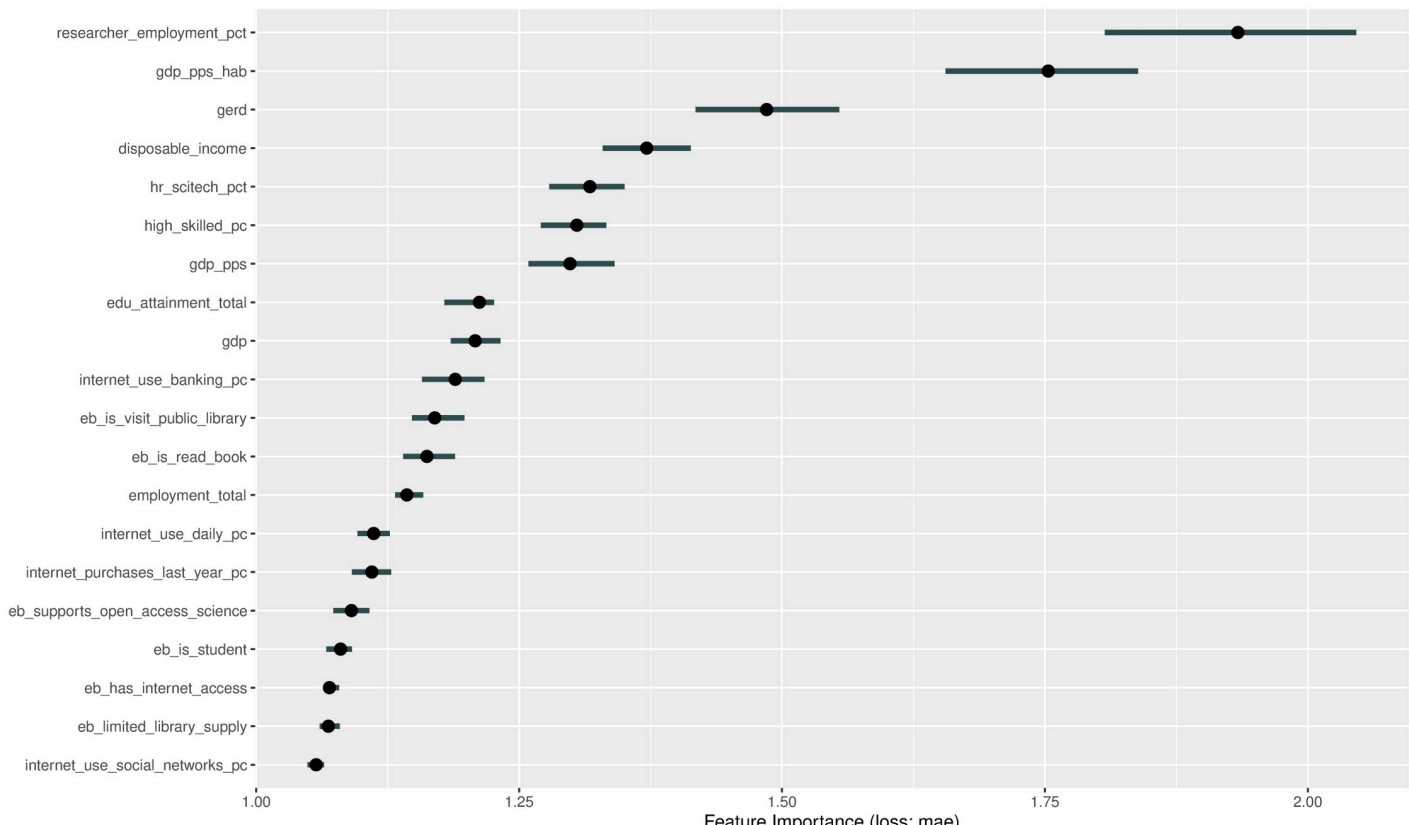

**Fig 6. Random forest feature importance of EUROSTAT+EUROBAROMETER (DV: Count per capita, number of runs: 100).**

**Table 10. European models IV: Eurobarometer variables (DVs: Download per capita, download per researcher).**

| | Model 16 DV:count per capita | Model 17a DV:count per researcher | Model 17b DV:count per researcher |
|---|---|---|---|
| (Intercept) | 6.204 *** | 8.160 *** | 6.905 *** |
| | (0.843) | (0.813) | (0.886) |
| log(GDP purchasing power parity) | 0.261** | 0.006 | 0.053 |
| | (0.080) | (0.078) | (0.082) |
| % of R&D personnel and researchers in the workforce | 0.673*** | | |
| | (0.056) | | |
| % of population who visited a public library at least once a year | -1.116** | -1.391** | |
| | (0.415) | (0.428) | |
| % of population who not visited public libraries more often because of perceived low-quality local supply | | | 3.963*** |
| | | | (0.886) |
| null.deviance | 2553172.101 | 350303.560 | 350303.560 |
| Deviance | 1061645.079 | 324682.437 | 312052.421 |

*** p < 0.001;

** p < 0.01;

* p < 0.05.

model (Model 17b), we found that the more people report not using a library due to its inadequate supply or resources, the higher the use of shadow libraries. While both these findings support our hypothesis that the quality of legal access infrastructures has a strong impact on shadow library usage, we treat these findings with some caution. The usefulness of these EUROBAROMETER variables is relatively limited due to the limited number of respondents, including the reliability of the statistics on a regional level.

In conclusion, both the 'brute force' approach, and the random forest approach was comprehensive in the way that it has measured all possible regression models, and the random forest comprehensively compared the predictive power of variables groups. Our inductive approach validated our initial theoretical framework, and revealed important potential library-related effects, that would need better data to fully confirm.

## Conclusions

In our earlier work on scholarly piracy, we conducted a supply side analysis. That research established that a significant chunk of the shadow library supply is not available in digital format and a significant share of downloads concentrate on legally inaccessible works. This offered a simplistic hypothesis: shadow library usage is mostly driven by market failures and the lack of convenient digital legal access alternatives.

In our present article, we offer a more detailed and elaborate picture on the piratical demand for scholarly work. Using comparable models to explain global differences in shadow library use on a country level and including a more fine-grained analysis of scholarly piracy within the EU, we arrive at similar conclusions.

Scholarly literature is a special information good. It is mainly used as an input for knowledge-intensive social and economic activities: (higher) education, and research and development. Its consumers are almost exclusively highly educated, possessing enough online proficiency to access often concealed shadow libraries. For the same reason, it can safely be assumed these consumers are aware of the legal and ethical dilemmas around the illicit access of copyrighted scholarly published materials.

We have found two significant demand drivers of scholarly piracy: GDP and the size of knowledge-intensive sector. Contrary to our initial, somewhat naïve assumption, we found that gross income and piracy is positively correlated. Free access piratical resources are used more often in high-income territories with potentially better legal access opportunities, such as libraries, and other institutional and individual access alternatives. This suggests that the lack of legal access infrastructures does not provide a satisfactory explanation for how shadow libraries are used.

In this article we have offered two alternative explanations. First, we have offered a model to differentiate the effect of income on knowledge demand at different levels of economic development. In our global models, we have shown that extra income has a much greater impact on shadow library demand in low-income countries than in high-income ones. This may be related to the mechanics of extra spending on knowledge intensive sectors. In low-income countries, extra spending increases piracy more as it expands the scope and amount of potential demand; while in high-income countries, extra spending may result relatively lower levels of piracy, because it results in better legal supply infrastructures, rather than the further expansion of demand.

Second, our European models suggest there are other, socioeconomic factors that limit the capacity to use and absorb freely accessible knowledge in the knowledge-intensive sectors of low-income regions. Even if the size of the knowledge-intensive sector is comparable to those in richer regions, less affluent regions face constraints that limit their ability to use and absorb

knowledge from freely accessible resources. That being said, we have found some evidence that points to the importance of good legal institutional access infrastructures: where libraries are used and found adequate, less scholarly piracy takes place.

These findings can also serve as a warning to the global open-access movement that is gaining momentum. Open access, legal or piratical, is hardly a panacea. As our study shows, access to knowledge is not the only, or most important constraint on knowledge-intensive social and economic activities at the peripheries. Access is only one aspect that defines the global dissemination and local use and usefulness of knowledge. A lot depends on the local conditions, which ultimately define to what extent freely accessible knowledge can be absorbed and utilized by both local individual and institutional actors.

This study has a number of limitations. The data it relies on is relatively dated. The geolocation of download data may be inaccurate due to a number of factors: the inaccuracy of IP address-to-geolocation dataset, our inability to fully detect and isolate clandestine traffic via VPNs, and automated traffic via bots and scrapers. We wish we had better datasets to separate different forms of demand: educational uses from university networks, R&D related demand by economic actors, and university research. Hopefully, we'll be able to address these issues in future work.

## Supporting information

**S1 File. Regional eurostat variables for understanding piracy of books.**
(PDF)

**S1 Table. Summary statistics for the variables in the global model.**
(PDF)

**S2 Table. European models (DV: Download counts).**
(PDF)

## Acknowledgments

The authors wish to thank the anonymous data donor for their generosity, and for consenting to the publication of the dataset.

## Author Contributions

**Conceptualization:** Balázs Bodó.

**Data curation:** Balázs Bodó, Dániel Antal.

**Formal analysis:** Balázs Bodó, Dániel Antal, Zoltán Puha.

**Funding acquisition:** Balázs Bodó.

**Investigation:** Balázs Bodó, Zoltán Puha.

**Methodology:** Balázs Bodó, Dániel Antal, Zoltán Puha.

**Project administration:** Balázs Bodó, Dániel Antal.

**Resources:** Balázs Bodó.

**Software:** Balázs Bodó, Dániel Antal, Zoltán Puha.

**Supervision:** Balázs Bodó.

**Validation:** Dániel Antal.

**Visualization:** Balázs Bodó.

**Writing – original draft:** Balázs Bodó.

**Writing – review & editing:** Balázs Bodó, Dániel Antal, Zoltán Puha.

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
