## [Decision Letter · Decision Letter 0]

17 Aug 2020

PONE-D-20-18408

Open access is not a panacea, even if it’s radical – an empirical study on the role of shadow libraries in closing the inequality of knowledge access

PLOS ONE

Dear Dr. Bodo,

Thank you for submitting your manuscript to PLOS ONE. After careful consideration, we feel that it has merit but does not fully meet PLOS ONE’s publication criteria as it currently stands. Therefore, we invite you to submit a revised version of the manuscript that addresses the points raised during the review process.

The peer reviewers have provided valuable advice to this article, and I'm agree with their judgement. For instance, the empirical statistics should be more accurate and standardized. Also ,the authors should pay more attention on the details according to the PLOS ONE’s publication criteria. 

We look forward to receiving your revised manuscript.

Kind regards,

Lizhi Xing, Ph.D

Academic Editor

PLOS ONE

Journal Requirements:

2. Please consider changing the title so as to meet our title format requirement (https://journals.plos.org/plosone/s/submission-guidelines). In particular, the title should be "Specific, descriptive, concise, and comprehensible to readers outside the field" and in this case it is not informative and specific about your study's scope and methodology.

"The research received funding from the H2020 Research grant #710722 "OPENing UP new methods, indicators and tools for peer review, dissemination of research results, and impact measurement", and was carried out on the Dutch national e-infrastructure with the support of SURF Cooperative."

We note that one or more of the authors are employed by a commercial company: CEEMID.

3.1. Please provide an amended Funding Statement declaring this commercial affiliation, as well as a statement regarding the Role of Funders in your study. If the funding organization did not play a role in the study design, data collection and analysis, decision to publish, or preparation of the manuscript and only provided financial support in the form of authors' salaries and/or research materials, please review your statements relating to the author contributions, and ensure you have specifically and accurately indicated the role(s) that these authors had in your study. You can update author roles in the Author Contributions section of the online submission form.

3.2. Please also provide an updated Competing Interests Statement declaring this commercial affiliation along with any other relevant declarations relating to employment, consultancy, patents, products in development, or marketed products, etc. 

5. We note that Figure 2 and 4 in your submission contain map images which may be copyrighted. All PLOS content is published under the Creative Commons Attribution License (CC BY 4.0), which means that the manuscript, images, and Supporting Information files will be freely available online, and any third party is permitted to access, download, copy, distribute, and use these materials in any way, even commercially, with proper attribution. For these reasons, we cannot publish previously copyrighted maps or satellite images created using proprietary data, such as Google software (Google Maps, Street View, and Earth). For more information, see our copyright guidelines: http://journals.plos.org/plosone/s/licenses-and-copyright.

5.1.    You may seek permission from the original copyright holder of Figure 2 and 4 to publish the content specifically under the CC BY 4.0 license.

5.2.    If you are unable to obtain permission from the original copyright holder to publish these figures under the CC BY 4.0 license or if the copyright holder’s requirements are incompatible with the CC BY 4.0 license, please either i) remove the figure or ii) supply a replacement figure that complies with the CC BY 4.0 license. Please check copyright information on all replacement figures and update the figure caption with source information. If applicable, please specify in the figure caption text when a figure is similar but not identical to the original image and is therefore for illustrative purposes only.

6. Please ensure that you refer to Figures 2, 4 and 5 in your text as, if accepted, production will need this reference to link the reader to the figures.

7. We note you have included tables to which you do not refer in the text of your manuscript. Please ensure that you refer to Tables 5, 7 and 10 in your text; if accepted, production will need this reference to link the reader to the Tables.

Reviewers' comments:

Reviewer's Responses to Questions

**Comments to the Author**

1. Is the manuscript technically sound, and do the data support the conclusions?

Reviewer #1: Yes

Reviewer #2: Yes

2. Has the statistical analysis been performed appropriately and rigorously? 

Reviewer #1: Yes

Reviewer #2: Yes

3. Have the authors made all data underlying the findings in their manuscript fully available?

Reviewer #1: Yes

Reviewer #2: Yes

4. Is the manuscript presented in an intelligible fashion and written in standard English?

Reviewer #1: Yes

Reviewer #2: Yes

5. Review Comments to the Author

Reviewer #1: The paper “Open access is not a panacea, even if it’s radical – an empirical study on the role of shadow libraries in closing the inequality of knowledge access” investigates on the economic and social factors behind the usage of shadow libraries, and provides interesting analysis in the global and European models. The publishers' subscription fee increases the financial burden of institutions and researchers. Shadow library was created to deal with complex institutional, political, financial, and other issues. However, academic piracy is worth studying, so this study is indeed a relevant and timely topic. This article is well organized and draws conclusions and arguments on the data analysis. I only have a few minor comments for the authors to consider, as to further improve the article:

1. In the “Data Overview and Descriptive Statistics” section, the author obtained many data sources, which provided the source and explanation of variables for the following model. It is important to show a table with the descriptive statistical information of variables in the model. This table can show the information of variables more intuitively and enhance readability.

2. In the construction of Global/European model, independent variables should use literature to support, especially for the authors to justify their research design (why some indicators were considered, but not the others)

3. In the Global Model, the authors write " Since our dependent variable is count data, the use of Poisson regression is justified. On the other hand, a negative binomial distribution did not suit this problem, therefore we omitted that approach " When the dependent variable is count data, Poisson regression and negative binomial regression are optional. Although Poisson regression is usually used, it needs explanation why negative binomial distribution is not suitable for this problem. If possible, the author should declare the reason for selecting regression methods, such as the relationship between expectation and variance.

4. There are three identical tables 3. I don't understand why three identical tables should be here.

5. In the Random forest models, I believe that the random forest model selected by the author has performed well in this study, but other model selection methods should also be considered. If possible, authors can add the comparison between other methods and the random forest model, which can better explain why the random forest method is used instead of others.

Reviewer #2: This is an illuminating and meaningful study. According to regression analysis in global models and European models, the authors found two significant demand drivers of scholarly piracy: GDP and the size of knowledge intensive sector; and revealed that open access knowledge might have limited usefulness in addressing knowledge access and production inequalities, in case of lacking inadequate or improper knowledge absorption infrastructures, which could be thought-provoking for the global open-access movement.

There are several minor problems that need to be addressed or improved.

1. The names of variables should be addressed clearly, such as “dl_per_pop_round”, “dl_per_pop”, “pop_per_mil”, “eb_is_visit_public_library” , “eb_limited_library_supply”, et al.

2. What is the statistical meaning of value in parentheses below the coefficient in each model? It doesn't look like the P value since the value doesn’t match the significant signal. Please give the necessary illustration.

3. Table 3 appears three times, and table 6 appears twice. Figure 6 is not shown in the manuscript. Please check the layout and details carefully.

4. In page 15 and 16, authors claimed the model 7 can “explain ~72% of the variance”, “effect of disposable income is positive and significant at a 95% level” in model 9, and “the download per researcher models have a worse fit than the per capita models”. How statistics support these statements? Could the certifiable statistics be available?

5. It seems not very clear to understand how the results of Random forest models connect with other models or support the research conclusion. Would the results of Random forest models be available and explained more specifically and clearly?

6. PLOS authors have the option to publish the peer review history of their article (what does this mean?). If published, this will include your full peer review and any attached files.

Reviewer #1: No

Reviewer #2: No

---

## [Author Response · Author response to Decision Letter 0]

10 Sep 2020

we responded to reviwer and editor comments in a standalon document. here we copy the contents of that document:

Rebuttal letter, manuscript ID#PONE-D-20-18408

Dear Reviewers, editors, 

First of all we would like to thank you for the useful reviews, and comments. Hopefully we addressed each point fully. This letter is thus less of a rebuttal, than a confirmation of us addressing each point as requested.

1. Please ensure that your manuscript meets PLOS ONE's style requirements, including those for file naming. We updated the document to conform with the requirements:

- changed the style of headings

- updated the formatting of references

- table titles have been moved on top of the tables

- figures have been renamed to Fig.

- tables have been converted to proper tables.

2. Please consider changing the title so as to meet our title format requirement.

We changed the title to Can scholarly pirate libraries bridge the knowledge access gap? An empirical study on the structural conditions of book piracy in global and European academia.

2. We note that one or more of the authors are employed by a commercial company. 

We would like to make a correction and requested statements for author Daniel Antal.

We would like to state that the Daniel Antal is an independent researcher. The affiliation information provided was confusing, and we would like to change it. CEEMID is not a commercial entity, as you have suggested in your remarks, but a trademark registered in the Dutch Chamber of Commerce to the sole proprietorship of Daniel Antal and a name of a collaborative project to pool open and other research data. This name and trademark was used as AD’s affiliation because the article uses some of the code and (open) data that CEEMID also uses. 

Daniel Antal is a sole proprietor under Dutch law, and he mainly provides data analysis. There was no employer, client or other funder that played any role in the study design, data collection and analysis, decision to publish, or preparation of the manuscript. Furthermore, he has no whatsoever conflicting interest, personal, professional or financial that is related anyhow to this publication, and his other activities as a sole proprietor do not alter our adherence to PLOS ONE policies on sharing data and materials. The only funding that Daniel Antal received for this work was paid from the identified funding source of this project, so no additional funding information is applicable.

Therefore, we believe that correct affiliation in this case is “independent researcher”.

4. We note that you have stated that you will provide repository information for your data at acceptance. 

We added the DOIs for both the data and the code to the paper.

Code and data repository: https://zenodo.org/record/4012352; DOI: 10.5281/ZENODO.4012352

Raw Data repository: https://uvaauas.figshare.com/projects/Shadow_Libraries/80837; DOI: 10.21942/uva.12330959

5. We note that Figure 2 and 4 in your submission contain map images which may be copyrighted. 

Both figures were created using open source software leaflet, and open access data by Openstreetmap. In both figures the copyright information is visible. Maps based on Openstreetmap data have been published before in PLOS ONE (for example: https://journals.plos.org/plosone/article/figure?id=10.1371/journal.pone.0165331.g001 and https://journals.plos.org/plosone/article/figures?id=10.1371/journal.pone.0209641) We updated the description to include the appropriate copyright information.

6. Please ensure that you refer to Figures 2, 4 and 5 in your text as, if accepted, production will need this reference to link the reader to the figures. DONE

7. Please ensure that you refer to Tables 5, 7 and 10 in your text; if accepted, production will need this reference to link the reader to the Tables. DONE

8. Please include captions for your Supporting Information files at the end of your manuscript, and update any in-text citations to match accordingly. Done.

Reviewers’ comments

1. In the “Data Overview and Descriptive Statistics” section, the author obtained many data sources, which provided the source and explanation of variables for the following model. It is important to show a table with the descriptive statistical information of variables in the model. This table can show the information of variables more intuitively and enhance readability.

For readability we included this table as supplementary material S2 Table.

2. In the construction of Global/European model, independent variables should use literature to support, especially for the authors to justify their research design (why some indicators were considered, but not the others)

We added extra language to the Data overview and descriptive statistics section to address this.

3. In the Global Model, the authors write " Since our dependent variable is count data, the use of Poisson regression is justified. On the other hand, a negative binomial distribution did not suit this problem, therefore we omitted that approach " When the dependent variable is count data, Poisson regression and negative binomial regression are optional. Although Poisson regression is usually used, it needs explanation why negative binomial distribution is not suitable for this problem. If possible, the author should declare the reason for selecting regression methods, such as the relationship between expectation and variance.

We added extra language and reference to justify our use of Poisson regression as opposed to neg binomial.

4. There are three identical tables 3. I don't understand why three identical tables should be here.

We could not find three identical tables in the submitted manuscript. 

5. In the Random forest models, I believe that the random forest model selected by the author has performed well in this study, but other model selection methods should also be considered. If possible, authors can add the comparison between other methods and the random forest model, which can better explain why the random forest method is used instead of others.

We added extra language to better explain other methods we used, and the role of random forest in the study.

Reviewer #2: 

1. The names of variables should be addressed clearly, such as “dl_per_pop_round”, “dl_per_pop”, “pop_per_mil”, “eb_is_visit_public_library” , “eb_limited_library_supply”, et al. Corrected.

2. What is the statistical meaning of value in parentheses below the coefficient in each model? Standard errors are in the parentheses. Added this info to the text.

3. Table 3 appears three times, and table 6 appears twice. Figure 6 is not shown in the manuscript. Please check the layout and details carefully.

As above, in the submitted manuscript we could not find these errors. 

4. In page 15 and 16, authors claimed the model 7 can “explain ~72% of the variance”, “effect of disposable income is positive and significant at a 95% level” in model 9, and “the download per researcher models have a worse fit than the per capita models”. How statistics support these statements? Could the certifiable statistics be available?

Added language. All statistics are available in the code repository.

5. It seems not very clear to understand how the results of Random forest models connect with other models or support the research conclusion. Would the results of Random forest models be available and explained more specifically and clearly?

Added extra language to explain better the use of random forest models.

---

## [Decision Letter · Decision Letter 1]

4 Nov 2020

Can scholarly pirate libraries bridge the knowledge access gap? An empirical study on the structural conditions of book piracy in global and European academia.

PONE-D-20-18408R1

Dear Dr. Bodo,

We’re pleased to inform you that your manuscript has been judged scientifically suitable for publication and will be formally accepted for publication once it meets all outstanding technical requirements.

Kind regards,

Sergi Lozano

Academic Editor

PLOS ONE

Additional Editor Comments (optional):

Reviewers' comments:

Reviewer's Responses to Questions

**Comments to the Author**

1. If the authors have adequately addressed your comments raised in a previous round of review and you feel that this manuscript is now acceptable for publication, you may indicate that here to bypass the “Comments to the Author” section, enter your conflict of interest statement in the “Confidential to Editor” section, and submit your "Accept" recommendation.

Reviewer #1: All comments have been addressed

Reviewer #2: (No Response)

2. Is the manuscript technically sound, and do the data support the conclusions?

Reviewer #1: Yes

Reviewer #2: (No Response)

3. Has the statistical analysis been performed appropriately and rigorously? 

Reviewer #1: Yes

Reviewer #2: (No Response)

4. Have the authors made all data underlying the findings in their manuscript fully available?

Reviewer #1: Yes

Reviewer #2: (No Response)

5. Is the manuscript presented in an intelligible fashion and written in standard English?

Reviewer #1: Yes

Reviewer #2: (No Response)

6. Review Comments to the Author

Reviewer #1: The author made good revision this time, so in my option, it can be published if the editors will accepted.

Reviewer #2: (No Response)

7. PLOS authors have the option to publish the peer review history of their article (what does this mean?). If published, this will include your full peer review and any attached files.

Reviewer #1: No

Reviewer #2: No

---

## [Editor Report · Acceptance letter]

10 Nov 2020

PONE-D-20-18408R1 

Can scholarly pirate libraries bridge the knowledge access gap? An empirical study on the structural conditions of book piracy in global and European academia. 

Dear Dr. Bodó:

I'm pleased to inform you that your manuscript has been deemed suitable for publication in PLOS ONE. Congratulations! Your manuscript is now with our production department. 

Kind regards, 

on behalf of

Dr. Sergi Lozano 

Academic Editor

PLOS ONE